# Effects of Microplastic Contamination on the Aquatic Plant *Lemna minuta* (Least Duckweed)

**DOI:** 10.3390/plants12010207

**Published:** 2023-01-03

**Authors:** Simona Ceschin, Flaminia Mariani, Dario Di Lernia, Iole Venditti, Emanuele Pelella, Maria Adelaide Iannelli

**Affiliations:** 1Department of Sciences, University of Roma Tre, Viale G. Marconi 446, 00146 Rome, Italy; 2Institute of Agricultural Biology and Biotechnology—National Research Council (IBBA-CNR), Via Salaria Km 29.300, Monterotondo Scalo, 00015 Rome, Italy

**Keywords:** poly(styrene-co-methylmethacrylate), free-floating plant, freshwater, microplastic adsorption, phytotoxic effect, chronic impact

## Abstract

Microplastics are widely spread in aquatic environments. Although they are considered among the most alarming contaminants, toxic effects on organisms are unclear, particularly on freshwater plants. In this study, the duckweed *Lemna minuta* was grown on different concentrations (50, 100 mg/L) of poly(styrene-co-methyl methacrylate) microplastics (MP) and exposure times (T0, T7, T14, T28 days). The phytotoxic effects of MP were investigated by analyzing several plant morphological and biochemical parameters (frond and root size, plant growth, chlorophyll, and malondialdehyde content). Observations by scanning electron microscope revealed MP adsorption on plant surfaces. Exposition to MP adversely affected plant growth and chlorophyll content with respect to both MP concentrations and exposure times. Conversely, malondialdehyde measurements did not indicate an alteration of oxidative lipid damage in plant tissue. The presence of MP induced root elongation when compared to the control plants. The effects of MP on *L. minuta* plants were more evident at T28. These results contribute to a better understanding of MP’s impact on aquatic plants and highlight that MP contamination manifests with chronic-type effects, which are thus detectable at longer exposure times of 7 days than those traditionally used in phytotoxicology tests on duckweeds.

## 1. Introduction

Plastics are synthetic organic polymers that are mainly derived from fossil fuel-based chemicals like natural gas or petroleum [1]. Certain characteristics of plastics have caused their wide use, such as their lightweight nature, versatility, strength, longevity, hygiene, food compatibility, and washability. Because of that, plastics production increased from 15 million tons in 1964 to more than 368 million tons produced in the year 2019, of which 114 were produced in China alone and 59 in Europe, and its capacity is expected to double by 2040 [2].

In short, plastics have been and continue to be one of the most widely used synthetic materials in the world, in daily life products as well as in agriculture and industry. Despite the benefits of using plastic products, the release of large amounts of this material into the environment has become a cause for increasing global concern, being considered the second most alarming environmental problem after global warming [2].

Unauthorized discharges and inadequate waste management lead to the massive release of plastics into the environment that accumulates in various environmental matrices, taking many years to degrade. In addition, plastic waste debris exposed to weathering gradually fragments into smaller pieces that are dispersed into aquatic and terrestrial environments [3,4], increasing their potential for adsorption, ingestion, and accumulation by living organisms [5].

The polymers most commonly produced as plastics are polystyrene (PS), polyurethane (PUR), polyvinyl chloride (PVC), polypropylene (PP), polyethylene (PE), polyethylene terephthalate (PET), and their copolymeric compounds, which together account for about 80 percent of total plastic production [6].

Plastics can be classified according to their size in: macroplastics (>25 mm), mesoplastics (5–25 mm), microplastics (0.1–5 mm), and nanoplastics (<100 nm) [2]. In aquatic environments, the main sources of microplastics are direct discharges from water treatment plants, industrial and agricultural wastewater, and spontaneous degradation of macro- and mesoplastics in water [7]. When microplastics enter the aquatic environment, some remain suspended in the body of water, others float to the surface, and still others with a higher density settle to the bottom. Animal organisms belonging to different trophic levels can easily ingest microplastics, and more and more cases are highlighting the worrying phenomenon of biomagnification of plastics along the food chain [7].

Although most of the research has been focused primarily on the ecological problem of plastic contamination in marine ecosystems [1,8,9,10,11], some recent investigations have highlighted that plastics are an equally serious source of environmental risk to freshwater ecosystems [12,13,14,15,16].

Furthermore, investigations of microplastics and their impact on aquatic communities have mainly focused on animal organisms, largely neglecting plant organisms, even though plants play a central role in ecosystems both trophically, structurally and functionally [17,18]. Furthermore, plants, the first interface between the abiotic and biotic components of an ecosystem, assume the important role of early warning systems, essential for early intercepting contamination and, therefore, limiting biomagnification processes both along the food chain and in the environment [19].

Although aquatic and bank plants are often exposed to plastic pollution, studies verifying the effects of microplastics on freshwater plants are still scarce and concern very few species [19], including microalgae of the genus *Scenedesmus* and *Chlorella* and some flowering plants, such as *Lemna minor* L. and *Myriophyllum spicatum* L. [18,20,21]. The available literature points out that the main phytotoxic effects of plastics are inhibition of photosynthesis and limitation of shoot and root growth. These effects would appear to be due to microplastic particles adsorbing on the outer plant tissues and forming physical blockages to light and air by hindering photosynthesis and respiration activities [7,20,22,23,24,25]. However, some of these studies showed that generally, plant species are only affected when the concentrations of microplastics are higher than those occurring in nature [20,24].

Low-density microplastics with specific densities < 1 g cm^−3^, such as microspheres of polyethylene (PE), polystyrene (PS), and polypropylene (PP), are distributed in the upper layers of slow-flowing waters [26,27]. Here, microplastics frequently encounter pleustophytes, that are aquatic plants free-floating on the water surface, which have roots and the lower surface of vegetative body in direct contact with the water. Very common and widespread pleustophytes are the duckweeds (Lemnaceae family) that, although characterized by a very tiny vegetative body (frond), are able to produce a floating plant mat [28], which can easily trap plastic material in slow water of lakes, ponds, canals, lentic stretches of rivers, or smaller water basins [22]. Some field observations have shown how some duckweed species of the *Lemna* genus, such as *Lemna minuta* Kunth, retain a large amount of surface floating pollutants, including microplastics. The scarcity of studies on the interactions and effects of microplastics on duckweeds makes it necessary to acquire more information about it, as the duckweeds play a major role in aquatic ecosystems. Indeed, they serve both as habitats for many animal species, providing protection from predators or sites for larval spawning, and as a food source for many insects, fish, and waterfowl [29], becoming the basis of many food chains in aquatic environments. Based on the above-mentioned characteristics, duckweed species could be used for the removal of plastic material from the aquatic environment by phytostabilization. Phytostabilization involves stabilizing and binding the contaminant by adsorption on the leaves and roots, reducing its dispersion in water. Recently, it has been shown that, in addition to dissolved contaminants [30] and nanoplastics [31], microplastics may be adsorbed on the surface and accumulated by vascular plants [20,32]. Although they have not yet been studied consistently, several mechanisms have been suggested to explain this phenomenon of the adsorption of plastic material by aquatic macrophytes. The electrostatic forces of the cellulosic constituents of plant cells can attract microplastics, and their adsorption is facilitated by the roughness of plant surfaces, which provide many binding sites for plastic particles [33]. The surface morphology of the plant organisms can also play an important role in microplastic-plant interactions; in fact, for micro- and macroalgae, the more complex the algal thallus structure is on the surface, the more it can trap microplastics [34,35,36]. In addition, if a periphyton layer (e.g., composed by microalgae) is present on the plant surfaces, it creates a higher viscosity that increases the retention of microplastics [37].

Polystyrene (PS) and their copolymeric compounds, due to their insulating properties and extreme lightness, are among the most widely used plastic materials in construction and beyond. They can be found on the market in the common version (hard and rigid) or in the form of an expanded product (commonly called polystyrene), depending on their functions. Expanded PS is mainly used for packaging, and thermal and electrical insulation, while common PS is used for many disposable items (e.g., cutlery, razors, CD cases), furniture items, tableware, toys, lining of household appliances, and for many other items. Especially, polymethyl methacrylate is widely used in medicine for bone cements, contact and intraocular lenses, screw fixation in bone, filler for bone cavities and skull defects, vertebral stabilization in osteoporotic patients, and for packaging of medical devices [38]. In this research, poly(styrene-co-methyl methacrylate) microplastics (MP) were analyzed due to their widespread use and the large amount of waste generated by this low-density material.

Namely, this study aimed to assess (i) the ability of the duckweed *L. minuta* to adsorb MP from the water medium and (ii) the phytotoxic effects of MP on this aquatic plant. The results obtained can contribute to better understanding the type of impact that MP may have on aquatic plant organisms as well as further investigating the adsorption mechanisms of these contaminants, knowledge of which can be relevant to safeguarding the health not only of the plant community but also of the entire aquatic ecosystem.

## 2. Results and Discussion

### 2.1. Water Chemical and Physical Parameters

Water chemical- and physical parameters were measured at T0, T7, T14, and T28 in the control tests and in two treatments with MP (MP50 and MP100) (Appendix A).

In all tests, the dissolved oxygen concentration (DO) of the water increased overall over time, but in both microplastic treatments the values were significantly higher than in the control (*p* < 0.001; increase of about 20%) (Figure 1; Appendix A). This result is justifiable by considering the positive correlation between DO and amount of Lemna biomass recorded during the experiment (rho = 0.855; p < 0.001) (Figure 2). In fact, in the MP50 and MP100 treatments, the amount of biomass, although increasing over time and positively correlated with DO (rho = 0.855; *p* < 0.001), was significantly lower at T28 (a decrease of about 45%) than in the control, forming thinner floating mats that did not cover the entire water surface, therefore, allowing gaseous exchanges between air and water. On the contrary, in the control test, the greater amount of biomass produced limited gas exchanges between air and water, causing a higher DO reduction compared to the treatments. The negative influence of *L. minuta* floating mats on DO concentrations as a function of their thickness confirms findings from previous laboratory and field studies on the impact of this duckweed on the water chemical and physical components in aquatic ecosystems [39,40].

The mean water temperature values in the two MP treatments did not differ significantly from the control (*p* > 0.05) (Appendix A), varying in function of the air temperature recorded in the laboratory. The mean pH values increased progressively throughout the experiment in all tests, increasing by more than one unit in both treatments compared to T0 (Appendix A). However, pH was significantly different only during MP100 treatment compared to the control (Appendix A).

Conductivity, a factor dependent on the water ionic components, and salinity, related to the dissolved salt content of the water, decreased in all tests from T0 to T7, due to the presence of the starting *Lemna* populations, which absorbed these solutes for their vegetative growth. However, from T7 to T28, conductivity and salinity showed a steady increase (Appendix A), which is most likely due to weekly refills of mineral water rich in ions and salts, whose utilization by *Lemna* plants failed to compensate for the continuous inputs.

### 2.2. Effects of Microplastics on Plant Growth Parameters

In general, the amount of biomass (FW and DW) increased linearly in all tests during the experiment. However, up to T14, no significant differences (*p* > 0.05) were observed between the tests, whereas at T28, the difference was highly significant (*p* < 0.001) between the control (an increase of 130%) and treatments, as was consequently the difference recorded for the relative growth rate (RGR) (*p* < 0.001) (Figure 3 and Figure 4; Appendix A). These results show that exposure of *L. minuta* populations to MP induces a negative effect on plant growth and that this effect occurs at both MP concentrations tested but after a prolonged period of exposure. These findings would differ from previous studies conducted on the related species *L. minor*, exposed for seven days to microplastics [20,22], in which no significant effects on biomass and RGR were recorded.

*Lemna* samples observed by SEM showed MP microspheres adsorbed on roots and both upper and lower frond surfaces (Figure 5). This suggests the hypothesis that the adsorption of microparticles on roots and frond surfaces physically impedes the smooth passage of light and oxygen and the uptake of nutrients, consequently limiting the regular growth of the plant.

Frond length and width and root width showed no significant differences between controls and treatments (*p* > 0.05) in the presence of MP. In contrast, root length increased significantly from T14 to T28 during exposure to the two different MP concentrations (MP50, 85%; MP100, 50%) compared with the control (Figure 6; Appendix A). Already, Kalčíková et al. [22] highlighted that depending on composition, concentration, and size, microplastics show different impacts on plant growth, such as a re-direction of growth between root and frond or between root thickness and root elongation. However, while they recorded that microplastics cause mechanical stress that hinders root growth, in contrast, root elongation occurred in this study. The increase in root length observed in both treatments might suggest that treated *Lemna* samples tend to elongate their roots to reach those portions of the water column furthest from the surface; here, in fact, there are fewer MP microparticles in suspension due to their low specific density [26,27], and thus the plant might have more surface area available to take up water and nutrients without physical obstructions.

### 2.3. Effects of MP on Biochemical Parameters

Total chlorophyll content (Chl_tot_) was not affected by the exposition to both MP concentrations (MP50 and MP100) until T7, and these results agree with other studies carried out on L. minor over shorter experimental times [20,21,22]. Differently, in the MP100 treatments both at T14 and T28, Chl_tot_ was significantly lower than in the control (*p* < 0.001) and the reduction was about 20% (Figure 7; Appendix A), while for MP50 the pigment content is slightly decreased 10% only at T28. Long-term exposure to high microplastic concentrations led to chlorosis effects on *L. minuta* plants, as it is observable in Figure 3. Conversely, some studies highlighted that microplastics had no significant effect on *L. minor* chlorophyll content over both shorter [20,21,22] and longer experimental times [21].

Malondialdehyde (MDA) content showed no changes during the experiment, and no significant difference was observed between the treatments MP50 and M100 compared to the control (Figure 7; Appendix A). MDA measurement is closely related to lipid peroxidation activity occurring at the subcellular level, and its increase implies the induction of an oxidative stress condition. Therefore, these results on MDA content suggest that the MP used were adsorbed, but not absorbed by *Lemna*, which then did not internalize them.

### 2.4. Adsorption of MP on Plants

Quali-quantitative SEM observation of single *Lemna* samples from the two treatments revealed that MP were primarily adsorbed on the portions of the plant most exposed to the contaminant, namely the abaxial surface of the fronds, secondarily, the adaxial surface (Figure 5). In addition to MP, the presence of many pinnate diatoms (unicellular microalgae with silicified plant walls) was noted, especially on the abaxial surfaces (Figure 5). Preliminary observations of the *Lemna* samples collected in the field highlighted that these microalgae were already associated with *Lemna* in nature, and, thus, their presence was not related to a later contamination occurring in the laboratory during the experiment. An aspect that relates diatoms and microplastics has been pointed out in some recent studies where plastic seems to behave as a “rigid” substrate on which microalgae can more easily adhere and grow according to the phenomenon of biofouling [41,42]. The cause of such aggregation is not yet clear, but specific properties of plastics may help microalgae aggregation and growth [43]. Adsorption of MP on *Lemna* samples may have facilitated increased adhesion of microalgae on the surfaces of the treated *Lemna* samples, potentially amplifying the phytotoxic effects of microplastics.

## 3. Materials and Methods

### 3.1. Production of Microplastics

Microplastics (MP) used to investigate the effect on *L. minuta* plants were obtained from pellets of poly(styrene-co-methyl methacrylate) [P(S-co-MMA)] (Aldrich 462896, pellets average Mw 100,000–150,000 pellets, styrene 40%) following the OBM method reported in a previous work [44]. In particular, 300 mg of P(S-co-MMA) were dissolved in 10 mL of acetone (C_3_H_6_O, technical grade, Merck) and stirred for 24 h, then an aliquot of 7 mL was transferred into a dialysis cellulose membrane (width 10 mm, Sigma Aldrich D9277-100FT) and further immersed into 200 mL of distilled water for 5 days at constant temperature (T = 24 °C). MP were observed using a Gemini 300 field emission SEM system (Carl Zeiss AG, Jena, Germany), and their mean diameter was verified on SEM images by ImageJ software vers. 1.53t (National Institutes of Health, Bethesda, MD, USA). The mean diameter ± SE was calculated to be 2.60 µm ± 1.54 (Figure 8).

Stock MP suspensions were prepared at two different concentrations, 50 (MP50) and 100 (MP100) mg/L. These concentrations, on average higher than those recorded in nature, were used to stress the system with the aim of obtaining more evident biological responses and comparing the effects with other similar studies [20,21,22].

### 3.2. Plant Material and Experimental Set-Up

Samples of *L. minuta* were collected from a natural pond within the Appia Antica Regional Park (Rome, Italy), and then transported to the laboratory in containers containing local water. *Lemna minuta* samples were acclimated for seven days in mineral water of known chemical composition (Appendix A). The mineral profile of the chosen water was the closest match to the one in ponds where *L. minuta* grows spontaneously in nature [28].

100 mL of MP stock suspensions were transferred into cylindrical glass containers (7 × 7 mm, 240 mL), and an amount of 0.6 g of *L. minuta* fronds was added, corresponding to about 80 percent coverage of the water surface. In parallel, a control set was arranged with *L. minuta* and water without MP. Three replicates (n = 3) were set up for the two MP concentrations (MP50, MP100) and control. Plants in the control and treatment tests were grown for 28 days and were sampled at different time points: 0 (T0), 7 (T7), 14 (T14), and 28 (T28) days (Figure 3). The choice of longer exposure times than those used in similar studies [20,21,22] is based on the assumption that microparticles mainly cause a chronic, rather than acute, toxic effect [45].

Every 7 days, to restore the water level to 100 mL, each container was refilled using the same water medium in which the plant samples were grown.

### 3.3. Determination of Water Chemical and Physical Parameters

The measurement of water chemical and physical parameters, such as temperature (T, °C), pH (pH, pH values), conductivity (C, µS/cm), salinity (S, ‰), and dissolved oxygen concentration (DO, mg/L) was done using a multiparameter immersion probe (Hach-Lange HQ40d) at different time points T0, T7, T14 and T28 days.

### 3.4. Plant Growth and Morphological Measurements

#### 3.4.1. Plant Growth Analysis

The biomass amount of *L. minuta* was measured at each experimental time (T0, T7, T14, and T28) to quantify any changes during plant growth. *Lemna minuta* biomass was collected with a fine-mesh metal sieve and, after having dried for one minute on blotting paper, fresh weight (FW) was measured using a precision scale (AS2001, Digital Scale, Ascher). Then, plant biomass was completely dried for 72 h at 60 °C to determine dry weight (DW). Thereafter, Relative Growth Rate (RGR, g^−1^day^−1^) was calculated using the following formula [46]:RGR = (ln DW_f_ − ln DW_i_)/(T_f_ − T_i_)
where: DW_f_ = final dry weight (g), DW_i_ = initial dry weight (g), T_f_ = total incubation period (day), and T_i_ = initial time (day) at each experimental time.

#### 3.4.2. Morphological Analysis

To examine possible variations in frond and root sizes, five individuals of *L. minuta* were taken from each replicate and placed on graph paper to be observed and photographed under a stereomicroscope (Stemi 305, ZEISS). Specifically, length and width of both fronds and roots were measured for each individual, using ImageJ software vers. 1.53t (National Institutes of Health, Bethesda, MD, USA).

### 3.5. Analysis of Biochemical Parameters

At each experimental time, biochemical parameters such as total chlorophyll (Chl_tot_) and malondialdehyde (MDA) content were measured to analyze the plant physiological performance in response to MP treatments.

#### 3.5.1. Determination of Chlorophyll Content

Total chlorophyll content in *Lemna* fronds was measured as an indicator of the physiological status of the plant. Fresh *Lemna* fronds (0.2–0.5 g) were soaked in 10 mL of 95% (*v*/*v*) ethanol for 3 days in a stoppered tube at room temperature and in the dark. Samples were centrifuged at 3000× *g* for 10 min, and the absorbance of the supernatant was measured at 663 and 645 nm [47]. Chlorophyll concentration was calculated following the equations described by Huang et al. [46]:Chl*_a_* = 12.72 A_663_
*−* 2.69 A_645_(1)
Chl*_b_* = 22.90 A_645_
*−* 4.68 A_663_(2)
Chl_Tot_ = Chl*_a_* + Chl*_b_*(3)
where Chl*_a_*, Chl*_b_*, and Chl_Tot_ represent chlorophyll *a*, chlorophyll *b*, and total chlorophyll contents, respectively. A_663_ and A_645_ are the absorbances at 663 and 645 nm, respectively. Results are expressed as mg of total chlorophyll per gram of fresh weight plant tissue (mg/g FW).

#### 3.5.2. Determination of Malondialdehyde Content

Lipid peroxidation was measured by spectrophotometric methods by estimating malondialdehyde (MDA) content, which is considered a biomarker of oxidative damage in plant tissue and thus an indicator of plant stress conditions.

Frozen samples were homogenized in a precooled mortar and pestle with two volumes of ice-cold 0.1% (*w*/*v*) trichloroacetic acid (TCA), and 1 mM Ethylenediamine tetraacetic acid (EDTA) and centrifuged for 15 min at 16,000× *g*. A mixture containing 1 mL of supernatant and 2 mL of 0.5% (*w*/*v*) thiobarbituric acid (TBA) in 20% (*w*/*v*) TCA was heated to 95 °C for 30 min and then rapidly cooled in an ice bath.

After centrifugation (16,000× *g* for 10 min at 4 °C), the absorbance of the supernatant was read at 532 nm, and the values corresponding to non-specific adsorption at 600 nm were subtracted. The concentration of MDA was calculated using the extinction coefficient (ε = 155 mM/cm).

### 3.6. SEM Observations of Microplastics

At the end of each experimental time (T0, T7, T14, and T28), from each control and treatment test, 50 fronds of *Lemna* were randomly taken and dehydrated through EtOH baths in series at increasing concentrations (10, 30, 50, 70, 90, and 100%). Then, they were dried at the critical point (Bal-Tec CPD 030), mounted on a stub (using self-adhesive carbon discs), gold sputter coated (Emitech k550), and observed by scanning electron microscope (SEM) (Gemini 300, Carl Zeiss AG, Jena, Germany).

On selected acquired SEM images, the adsorption of MP on *L. minuta* was verified considering all plant surfaces, thus both the adaxial and abaxial surfaces of the frond as well as the entire root surface.

### 3.7. Statistical Analyses

Multiple two-way ANOVA tests were conducted to compare changes in chemical and physical water parameters and plant physiological measurements in the separate tests (C, MP50, MP100) and over different exposure times (T0, T7, T14, e T28). A post-hoc analysis (Tukey’s Test) was conducted for each ANOVA test. Assumptions of normality and homoscedasticity were tested both prior to ANOVA and on the model residuals. The correlation between biotic and abiotic parameters that were significantly different between treatments was investigated by calculating Spearman’s rank correlation coefficient. Where there was a strong and significant correlation, further analyses were conducted via analysis of covariance (ANCOVA). Where assumptions of homoskedasticity were not met, a logarithmic transformation of the dependent variable was performed. Non-significant interaction terms were removed from the ANCOVA models via stepwise selection. Graphs were made using the ggplot2 and sjPlot packages [48,49]. All statistical analyses were conducted using R software vers. 4.2.1 [50].

## 4. Conclusions

This study demonstrates that the aquatic plant *L. minuta* can adsorb MP large 1–5 μm, whose adsorption occurs mainly on plant surfaces in direct contact with the contaminated suspension (i.e., abaxial frond surface). The amount of adsorbed MP on *L. minuta* fronds was dose- and time-dependent. Indeed, MP most affected *L. minuta* growth (biomass, RGR) at the highest concentration (MP100) and after 28 days of exposure (T28). Simultaneously, reduction of chlorophyll content was evident, indicating that the plant exposition to MP contamination has reduced its photosynthetic capacity. Anyway, long-term monitoring of the effects of MP on the growth and biochemical parameters of *L. minuta* pointed out that this plant can tolerate high MP concentrations. Thus, free-floating mats of *L. minuta*, thanks to that tolerance and ability of phytostabilizing MP particles, it could be exploited in the phytoremediation of water contaminated by microplastics. It should be noted that the ability of this plant to capture microplastics can depend by different environmental conditions; for example, the natural presence of periphyton on the plant tissue may increase the number of microplastics adsorbed by the plants.

As a whole, these results contribute to a better understanding of microplastics impact on aquatic plants and highlight that MP contamination manifests with chronic-type effects, thus observable at longer exposure times than those traditionally used of 7 days in phytotoxicity tests on duckweeds [51,52]. Further investigations on the adsorption mechanisms of microplastics by this duckweed will be relevant to verifying the actual possibility of using this type of plant organisms in the phytoremediation of freshwaters contaminated with microplastics, which would currently seem to be one of the most promising biological approaches to remove microparticles in situ and then to safeguard the health of the plant community and the entire aquatic ecosystem.

## Figures and Tables

**Figure 1 plants-12-00207-f001:**
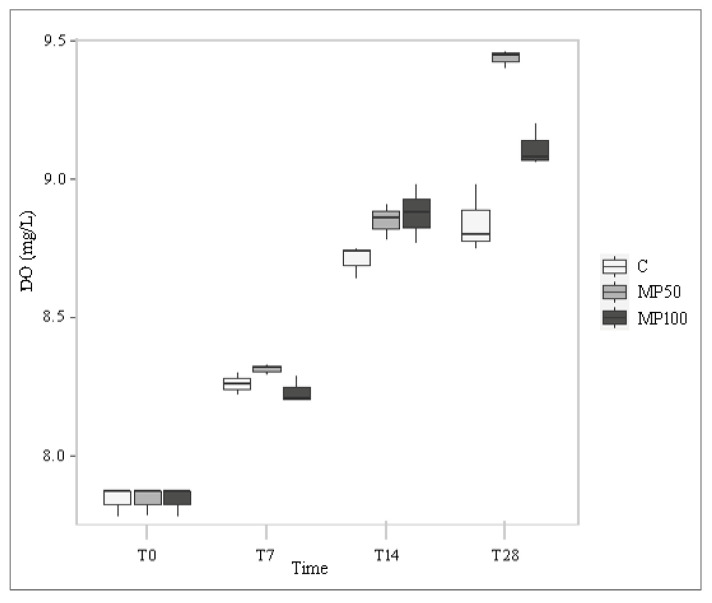
Dissolved oxygen (DO) in the control (C) and microplastic treatments (MP50 and MP100) at different exposure times (T0, T7, T14, and T28).

**Figure 2 plants-12-00207-f002:**
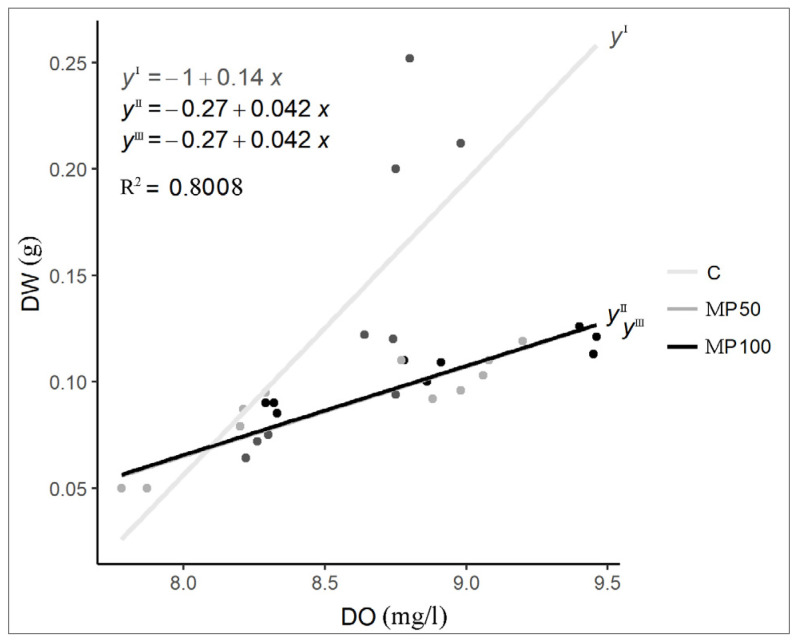
Regression model between DW and DO in control and microplastic treatments (MP50 and MP100). The regression lines of MP50 and MP100 are overlapping.

**Figure 3 plants-12-00207-f003:**
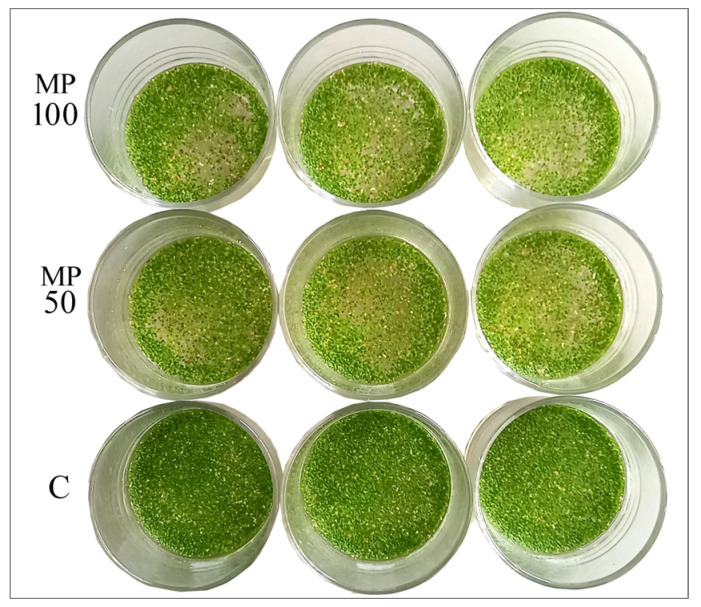
Biomass of *L. minuta* in the control (C) and microplastic treatments (MP50 and MP100) at T28. For each group, the three replications are shown.

**Figure 4 plants-12-00207-f004:**
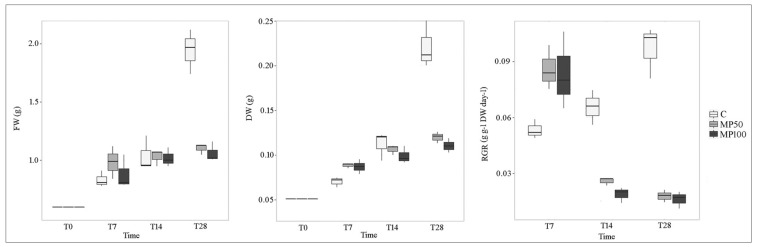
Fresh weight (FW), dry weight (DW), and relative growth rate (RGR) of *L. minuta* in control (C) and microplastic treatments (MP50 and MP100) at different exposure times (T0, T7, T14, and T28).

**Figure 5 plants-12-00207-f005:**
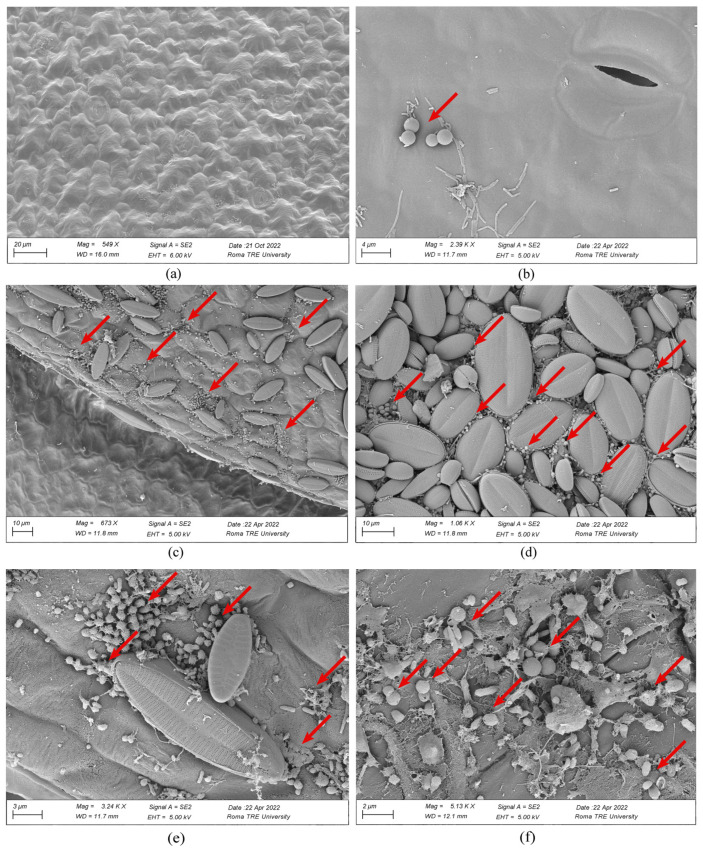
Scanning Electron Microscopy (SEM) images of *L. minuta* fronds grown in aqueous medium without (**a**) and with MP particles (arrows) (**b**–**f**). In detail: adaxial frond surface in control test at T28 (**a**), MP microspheres adsorbed onto the adaxial surface in PS50 treatment at T7 (**b**), and on abaxial surfaces in PS50 at T7 (**c**), in PS50 at T28 (**d**), and in PS100 at T28 (**e**–**f**). Evident aggregates of MPs with pennate diatoms (**c**–**e**).

**Figure 6 plants-12-00207-f006:**
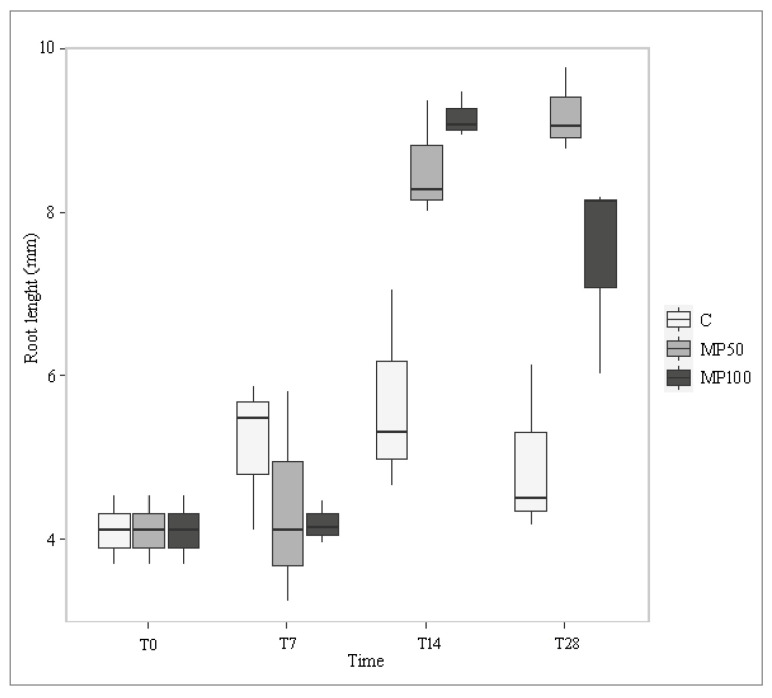
Root length (mm) of *L. minuta*, measured in control (C) and microplastic treatments (MP50, MP100) at different exposure times (T0, T7, T14, T28).

**Figure 7 plants-12-00207-f007:**
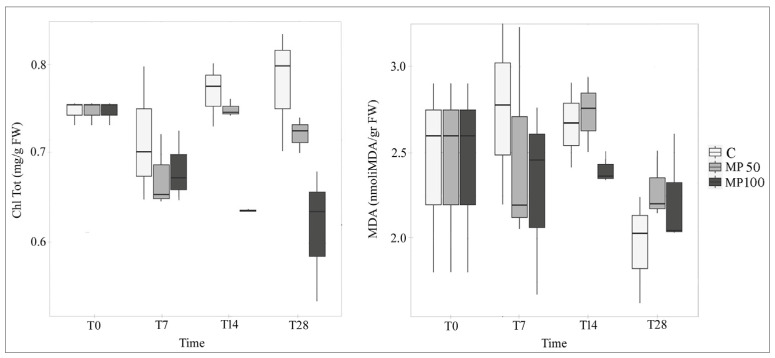
Total chlorophyll (Chl_tot_) and malondialdehyde (MDA) content in *L. minuta* samples in control (C) and microplastic treatments (MP50, MP100) at different exposure times (T0, T7, T14, T28).

**Figure 8 plants-12-00207-f008:**
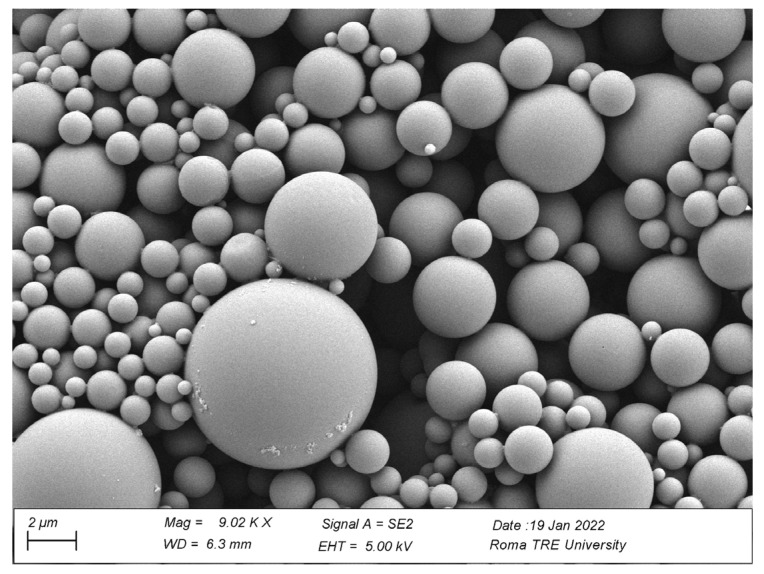
Scanning electron microscope (SEM) image of stock suspension of MP particles.

## Data Availability

Not applicable.

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
