# Peer review of "Effects of Microplastic Contamination on the Aquatic Plant Lemna minuta (Least Duckweed)"

_plants, 2023, doi:10.3390/plants12010207_

Round 1

Reviewer 1 Report

The authors performed a valuable and detailed study on effects of microplastic contamination on the aquatic environment. The experimental design, measurements and calculation were done correctly, and presentation of results is good. The main achievement is confirmation that Lemna minuta can adsorb and tolerate high microplastic contamination in the waterbody, and this ability can be useful as the possible phytoremediation strategy of freshwater contaminated with microplastic, which is currently among the most negative global threat for marine and freshwater ecosystems. The Reviewer did not found any methodological or professional errors in the manuscript, and the authors followed the Instructions for Authors when drafted the manuscript.

Author Response

Authors: We thank the Reviewer for these positive comments. No changes to the manuscript were requested by the Reviewer.

Reviewer 2 Report

This study investigated the phytotoxicity of microplastics. The scientific issues addressed are obvious and the research is significant. This study is worthy of publication in the journal, but the following issues still need to be noted before publication.

1.       In this study, MDA was selected as a biomarker, please give reasons for this choice. Why not consider other markers of enzymatic activity that may cause oxidative stress response, such as SOD, CAT, etc.

2.       Figure 2: Please clarify the meaning of each graph specifically in the figure or caption, for different treatment groups, is each graph a duplicate treatment group?

3.       Figure 4: It is suggested that the functional form of the fitted curve in the graph be given. Please also give the evaluation parameter corresponding to the fitting results, such as R2 or slope values, in order to distinguish the differences in the fitting effect between the different treatment groups.

4.       Figure 7: Please specify in the figure captions which treatment group is referred to in each figure. Also, which graph corresponds to the control group.

5. The authors refer to the impact of microplastic agglomeration issues on phytotoxicity. This requires the authors to extend their data on the physicochemical characterisation of microplastics, such as surface charge, particle size, etc. In this way, the relationship between agglomeration of microplastics and phytotoxicity can be explored quantitatively.

Author Response

1. Reviewer 2: In this study, MDA was selected as a biomarker, please give reasons for this choice. Why not consider other markers of enzymatic activity that may cause oxidative stress response, such as SOD, CAT, etc.

Authors: Thank you for this observation. We did not measure other oxidative stress marker since MDA is already a widely biomarker of oxidative damage in plant tissue and thus an indicator of plant stress condition. In addition, we thought that analyzing only MDA was enough since the results did not show any oxidative damage (MDA data - MDA even decreased at T28, i.e., at the end of the 'experiment) and some physiological parameters (biomass and pigments) changed due to prolonged exposure to MPs, probably due to physical impediments to plant growth.  Thus, probably, if we had different responses, such as an increase in MDA over time, it would have been useful to also analyze other biomarkers to support a state of oxidative stress in MPs-induced Lemna fronds.

2. Reviewer 2: Figure 2: Please clarify the meaning of each graph specifically in the figure or caption, for different treatment groups, is each graph a duplicate treatment group?

Authors: we clarified better this in the caption of the Figure 2.

3. Reviewer 2: Figure 4. It is suggested that the functional form of the fitted curve in the graph be given. Please also give the evaluation parameter corresponding to the fitting results, such as R2 or slope values, in order to distinguish the differences in the fitting effect between the different treatment groups.

Authors: We proposed a clearer and more precise visualization of the data in the graph of Figure 4 where, according to what was required, we added functional forms (equations) and R2 value (see Figure 4).

4. Reviewer 2: Figure 7: Please specify in the figure captions which treatment group is referred to in each figure. Also, which graph corresponds to the control group.

Authors: we specified in the figure caption which control or treatment group is referred to in each image. We have also added image corresponds to the control group since was missing in the previous version of the Figure 7.

5. Reviewer 2: The authors refer to the impact of microplastic agglomeration issues on phytotoxicity. This requires the authors to extend their data on the physicochemical characterization of microplastics, such as surface charge, particle size, etc. In this way, the relationship between agglomeration of microplastics and phytotoxicity can be explored quantitatively.

Authors: We thank the reviewer for this consideration. Surely it would be an interesting study to carry out the measurements of hydrodynamic radius and Z potential in the aqueous medium used for the experiment, but at the moment there is no possibility to make them. On the other hand, characterization by scanning electron microscopy (SEM) is also a suitable method to derive the average size and polydispersity of the polymeric nanoparticles, and we therefore used this technique, which was available to us.